# Evaluation of weight retention four weeks after delivery as a risk factor for gestational diabetes mellitus in a subsequent pregnancy

**Satoshi Shinohara●\*, Atsuhito Amemiya, Motoi Takizawa**

Department of Obstetrics and Gynecology, National Hospital Organization Kofu National Hospital, Kofu, Yamanashi, Japan

\* shinohara617@gmail.com

**Data Availability Statement:** All relevant data are within the manuscript and its Supporting Information files.

## Abstract

### Aim

We aimed to assess the association between postpartum weight retention (PPWR) in the fourth week after delivery and the risk of gestational diabetes mellitus (GDM) in a subsequent pregnancy.

### Methods

We performed a retrospective cohort study of the obstetric records of women who gave birth to their second singleton between 32 and 41 weeks of gestation at the National Hospital Organization Kofu National Hospital between January 2013 and September 2019. The exclusion criteria were missing data, twin pregnancy, diabetes in pregnancy, and delivery before 22 weeks in the first pregnancy. We calculated PPWR as the BMI 4 weeks after the first birth minus the BMI before the first pregnancy and grouped the subjects into the stable PPWR (gain of <1 BMI unit) and non-stable PPWR groups (gain of ≥1 BMI units). We used the $\chi^2$ test and multivariable logistic regression analysis to investigate the association between weight retention at the postpartum checkup and GDM.

### Results

We included 566 women in this study (mean age, 31.7±4.8 years; mean maternal pre-pregnancy BMI, 21.3±3.5 kg/m²; term delivery, n = 544 [96.1%]). The overall prevalence of GDM during the second pregnancy was 7.4% (42/566), and 33.9% (192/566) of women had stable PPWR. Non-stable PPWR was not significantly associated with GDM in the second pregnancy (adjusted odds ratio, 1.93; 95% confidence interval, 0.84–4.46) after controlling for each variable.

### Conclusion

PPWR measured in the fourth week after delivery was not associated with an increased risk of GDM in the second pregnancy.

**Funding:** The authors received no specific funding for this work.

**Competing interests:** The authors have declared that no competing interests exist.

## Introduction

Gestational diabetes mellitus (GDM) is a common pregnancy complication characterized by decreased insulin sensitivity and inadequate insulin response [1,2]. Research estimates indicate that GDM affects approximately 7–10% of all pregnancies worldwide [3–6]. GDM presents significant risks to the fetus, including an increased risk of macrosomia, neonatal hypoglycemia, hyperbilirubinemia, shoulder dystocia, and birth trauma [7,8]. Further, previous studies demonstrated that women with GDM have an increased risk of hypertensive disorders of pregnancy [9,10] and metabolic syndrome, type 2 diabetes mellitus, and cardiovascular disease later in life [11,12]. Therefore, it is important to identify the risk factors of GDM and establish effective and appropriate prevention strategies that improve maternal–fetal outcomes.

Previous studies showed that a history of GDM, a family history of diabetes, advanced maternal age, pre-pregnancy body mass index (BMI) changes between the first and second pregnancies, and maternal obesity are risk factors for GDM [13–16]. However, it is meaningless to inform a woman with these factors that she has an increased risk of GDM when she has already become pregnant again; physicians need to identify women with a high risk of GDM and implement primary prevention strategies before that point. Recently, Liu et al. reported that the findings of their retrospective cohort study suggest that excessive postpartum weight retention (PPWR) six weeks after the first pregnancy is a risk factor for GDM in the second pregnancy [8]. Their study is different than previous investigations in proposing that the postpartum checkup might provide the opportunity to identify factors that increase the risk of GDM in a subsequent pregnancy, allowing the implementation of primary prevention strategies. The postpartum checkup is important for obstetricians and other medical providers because, after this evaluation, most women will be lost to follow-up until they become pregnant again. Doctors could identify women with a high risk of not only GDM but also type 2 diabetes mellitus, and cardiovascular disease who need close monitoring in later life during the postpartum care period.

To our knowledge, Liu et al.'s study is the only one to address the relationship between PPWR at the postpartum checkup and GDM in the next pregnancy [8]. Further, GDM prevalence differs by race and ethnicity, and GDM risk factors appear to affect the disease differently across racial and ethnic groups [17]. Therefore, further studies are needed to clarify the potential impact of PPWR at the time of the postpartum checkup on glucose metabolism during the next pregnancy. We aimed to evaluate the association between PPWR four weeks after delivery of the first child and the risk of developing GDM during the second pregnancy in a Japanese cohort.

## Methods

### Study design

We performed a retrospective cohort study of women who gave birth to their second singleton between 32 and 41 weeks of gestation at the National Hospital Organization Kofu National Hospital, a community hospital, between January 2013 and September 2019. The exclusion criteria were missing data, twin pregnancy, diabetes in pregnancy, and delivery before 22 weeks in the first pregnancy. The National Hospital Organization Kofu National Hospital Human Subjects Review Committee approved the study protocol and waived the need for informed consent because of the retrospective study design. However, patients could refuse the use of their data through the hospital's website. All procedures were performed in accordance with the 1964 Helsinki Declaration and its later amendments.

## Data collection

We collected baseline demographic data and medical and family histories. We also obtained any history of diabetes in second-degree relatives from each patient. The obstetric records included maternal age at delivery, parity, pre-pregnancy BMI, pre-pregnancy BMI change between the first and second pregnancies, and the postpartum checkup BMI. PPWR was the exposure of interest. Because postpartum checkups are performed approximately four weeks after delivery in Japan [16], the PPWR at the checkup after the first pregnancy was defined as the BMI four weeks after the first birth minus the pre-pregnancy BMI of the first pregnancy. The pre-pregnancy weight for calculation of the BMI at the first visit was self-reported. We classified women according to their postpartum checkup PPWR and defined a stable PPWR as ≤1 BMI unit and non-stable PPWR >1 BMI unit, per previous relevant publications [8]. Moreover, advanced maternal age (≥35 years), pre-pregnancy obesity (BMI >25.0 kg/m$^2$), family history of diabetes, the interpregnancy BMI change, and previous GDM were used as explanatory variables because they have been previously described as GDM risk factors [13–16]. In Japan, an initial screening test followed by the diagnostic oral glucose tolerance test (OGTT) in screen-positive women, known as the two-step approach, is used for GDM diagnosis [16]. Moreover, the Japan Society of Obstetrics and Gynaecology has recommended the 50 g glucose challenge test (GCT) as a screening method for GDM [16]. Women with abnormal GCT results (serum glucose levels ≥140 mg/dL) subsequently undergo a 75-g OGTT for a definitive diagnosis. We diagnosed GDM if the patient had at least one abnormal plasma glucose value (≥92, ≥180, and ≥153 mg/dL for fasting, 1-hour, and 2-hour plasma glucose concentrations, respectively) after a 75-g oral glucose tolerance test [16].

Each patient's pre-pregnancy BMI was calculated according to the World Health Organization standard (body weight [kg]/height [m]$^2$) and recorded in her prenatal records at the first visit. We defined the interpregnancy BMI change as the difference between the pre-pregnancy BMI of the first and second pregnancies [15]. We classified women according to their interpregnancy BMI and defined a stable interpregnancy BMI as ≤1 BMI unit according to several relevant publications [8,15].

We did not evaluate maternal weight gain during pregnancy, a potential contributor to GDM [18], because we wanted to examine GDM risk factors before and early in the second pregnancy, and there is no consensus opinion regarding optimal weight gain during pregnancy, especially in obese women in Japan [16].

## Statistical analyses

We compared maternal and neonatal outcomes using Mann-Whitney U and $\chi^2$ tests. Fisher's exact test was used when the expected frequency was <5. We then performed a multivariable logistic regression analysis to examine the association between postpartum weight retention at the 4-week checkup and GDM in the second pregnancy, while controlling for potential confounding factors. All analyses were performed using Bell Curve for Excel (Social Survey Research Information Co., Ltd, Tokyo, Japan) and IBM SPSS Statistics for Windows, v. 25 (IBM Corp., Armonk, NY, US). The significance level was set at p<0.05.

## Results

During the study period, 620 women gave birth to their second singleton. After excluding 54 patients (missing data, n = 25; twin pregnancy, n = 11; delivered first singleton before 22 weeks of gestation, n = 16; diabetes in pregnancy, n = 2), we included 566 women who delivered between 32 and 41 weeks of gestation (mean age, 31.7±4.8 years). The mean maternal first and second pregnancy, pre-pregnancy BMIs were 20.8±3.3 kg/m$^2$ and 21.3±3.5 kg/m$^2$,

respectively. There were 544 (96.1%) term and 91 (16.1%) cesarean second singleton deliveries. The characteristics of women with a stable (≤1 BMI unit) and a non-stable PPWR (>1 BMI unit) were similar except for greater first and second pregnancy, pre-pregnancy BMIs, and lower first pregnancy, postpartum checkup BMIs in the stable PPWR group (Table 1).

The overall prevalence of GDM during the second pregnancy was 7.4% (42/566), 33.9% (192/566) of women had stable weight retention, and GDM was not significantly more common in the non-stable PPWR group (8.0% vs. 6.3%, p = 0.45; odds ratio [OR], 1.30; 95% confidence interval [CI], 0.65–2.61) (Table 2). Additionally, the interpregnancy BMI change, a history of previous GDM, and obesity were significantly associated with GDM in the second pregnancy (Table 2). Moreover, after controlling for each variable, the interpregnancy BMI change, a history of previous GDM, and obesity were significantly associated with GDM in the second pregnancy (non-stable interpregnancy BMI: adjusted OR, 2.11; 95% CI, 1.01–4.43; history of previous GDM: adjusted OR, 7.93; 95% CI, 3.15–20.0; obesity: adjusted OR, 4.23; 95% CI, 1.95–9.91) in the multivariable logistic regression analysis. However, non-stable PPWR was not significantly associated with GDM in the second pregnancy (adjusted OR, 1.93; 95% CI, 0.84–4.46). Finally, a multiple logistic regression model, into which we entered PPWR as a continuous variable, was performed. After controlling for each variable, a history of previous GDM, and obesity were significantly associated with GDM in the second pregnancy (history of previous GDM: adjusted OR, 7.70; 95% CI, 3.08–19.2; obesity: adjusted OR, 4.13; 95% CI, 1.91–8.95) in the multivariable logistic regression analysis. PPWR was not significantly associated with GDM in the second pregnancy (adjusted OR, 1.19; 95% CI, 0.94–1.50) (Table 3).

## Discussion

We evaluated PPWR in women delivering at 32–41 weeks of gestation at the National Hospital Organization Kofu National Hospital and found no significant association between the PPWR at the postpartum checkup after the first pregnancy and GDM in the second pregnancy. The incidence of GDM in our study was consistent with previous data [19,20].

The mechanism underlying the significant relationship between PPWR and GDM is still unknown. Liu et al., the first to report the association between weight retention at six weeks postpartum and subsequent GDM, did not describe the detailed mechanism of this relationship [8]. Instead, the team only cited previous studies that showed excessive postpartum weight retention is an important contributor to future maternal obesity [21,22], an important risk

**Table 1. Clinical characteristics in groups based on postpartum weight retention four weeks after the second pregnancy.**

|  | Stable PPWR n = 192 | Non-stable PPWR n = 374 | p-value |
|---|---|---|---|
| Maternal age | 32.1±5.0 | 31.4±4.7 | 0.09 |
| Premature delivery | 9 (4.7) | 13 (3.5) | 0.48 |
| Caesarean section | 25 (13.0) | 66 (17.6) | 0.16 |
| Instrumental delivery | 1 (0.50) | 2 (0.50) | 1.00 |
| First pregnancy, gestational length of pregnancy | 39.1±1.4 | 39.2±1.4 | 0.14 |
| First pregnancy, pre-pregnancy BMI | 22.1±4.2 | 20.1±2.4 | <0.001 |
| Second pregnancy, pre-pregnancy BMI | 21.9±4.2 | 21.0±3.1 | 0.02 |
| First pregnancy, postpartum checkup BMI | 22.0±3.8 | 22.3±2.7 | 0.003 |
| GDM | 12 (6.3) | 30 (8.0) | 0.45 |

Values are presented as mean±standard deviation or numbers (%).

PPWR, postpartum weight retention; GDM, gestational diabetes mellitus; BMI, body mass index

**Table 2. Crude and adjusted odds ratios of maternal risk factors for gestational diabetes mellitus in the second pregnancy.**

| Variable | GDM, (n) | Non-GDM, (n) | Crude | | Adjusted | | |
|---|---|---|---|---|---|---|---|
| | | | OR | 95% CI | OR | 95% CI | p-value |
| Postpartum checkup PPWR | | | | | | | |
| <1 BMI unit | 12 | 180 | 1 | Reference | 1 | Reference | |
| ≥1 BMI unit | 30 | 344 | 1.30 | 0.65–2.61 | 1.93 | 0.84–4.46 | 0.123 |
| Interpregnancy BMI change | | | | | | | |
| <1 BMI unit | 19 | 384 | 1 | Reference | 1 | Reference | |
| ≥1 BMI unit | 23 | 140 | 3.32 | 1.75–6.28 | 2.11 | 1.01–4.43 | 0.048 |
| Maternal age | | | | | | | |
| <35 | 26 | 374 | 1 | Reference | 1 | Reference | |
| ≥35 | 16 | 150 | 1.53 | 0.80–2.94 | 1.41 | 0.69–2.90 | 0.346 |
| Family history of diabetes | | | | | | | |
| No | 27 | 387 | 1 | Reference | 1 | Reference | |
| Yes | 15 | 137 | 1.56 | 0.81–3.04 | 1.27 | 0.61–2.63 | 0.524 |
| History of previous GDM | | | | | | | |
| No | 31 | 501 | 1 | Reference | 1 | Reference | |
| Yes | 11 | 23 | 7.73 | 3.46–17.3 | 7.93 | 3.15–20.0 | <0.001 |
| Obesity | | | | | | | |
| No | 25 | 470 | 1 | Reference | 1 | Reference | |
| Yes | 17 | 54 | 5.92 | 3.01–11.7 | 4.23 | 1.95–9.19 | <0.001 |

Values are presented as mean±standard deviation or numbers (%).

PPWR, postpartum weight retention; GDM, gestational diabetes mellitus; BMI, body mass index

PPWR was entered PPWR as a categorical variables.

**Table 3. Crude and adjusted odds ratios of maternal risk factors for gestational diabetes mellitus in the second pregnancy.**

| Variable | GDM (n) | Non-GDM (n) | Crude | | Adjusted | | |
|---|---|---|---|---|---|---|---|
| | | | OR | 95% CI | OR | 95% CI | p-value |
| Postpartum checkup PPWR in BMI unit median (25th–75th percentile) | 1.72 (0.74–2.68) | 1.49 (0.61–2.37) | | | 1.18 | 0.94–1.50 | 0.150 |
| Interpregnancy BMI change | | | | | | | |
| <1 BMI unit | 19 | 384 | 1 | Reference | 1 | Reference | |
| ≥1 BMI unit | 23 | 140 | 3.32 | 1.75–6.28 | 2.00 | 0.91–4.33 | 0.081 |
| Maternal age | | | | | | | |
| <35 | 26 | 374 | 1 | Reference | 1 | Reference | |
| ≥35 | 16 | 150 | 1.53 | 0.80–2.94 | 1.48 | 0.71–3.10 | 0.290 |
| Family history of diabetes | | | | | | | |
| No | 27 | 387 | 1 | Reference | 1 | Reference | |
| Yes | 15 | 137 | 1.56 | 0.81–3.04 | 1.34 | 0.64–2.78 | 0.440 |
| History of previous GDM | | | | | | | |
| No | 31 | 501 | 1 | Reference | 1 | Reference | |
| Yes | 11 | 23 | 7.73 | 3.46–17.3 | 7.69 | 3.08–19.2 | <0.001 |
| Obesity | | | | | | | |
| No | 25 | 470 | 1 | Reference | 1 | Reference | |
| Yes | 17 | 54 | 5.92 | 3.01–11.7 | 4.13 | 1.91–8.95 | <0.001 |

Values are presented as mean±standard deviation or numbers (%).

PPWR, postpartum weight retention; GDM, gestational diabetes mellitus; BMI, body mass index

PPWR was entered PPWR as a continuous variables

factor for GDM, as the basis of their study results [8]. There is an adaptive physiologic decrease in insulin sensitivity during pregnancy that is necessary for supplying energy sufficient to support fetal development [8]. During pregnancy, insulin sensitivity decreases by 50–60% under normal physiological circumstances [23,24]. However, overweight and obese individuals are typically less insulin sensitive than the normal-weight population [8,24]. Therefore, obese women tend to develop GDM, and blood glucose control is more difficult.

There are two likely explanations for the fact that, unlike previous investigators, we did not find a significant relationship between PPWR and GDM. Firstly, it is widely accepted that six weeks after delivery [8,25,26], most of the changes associated with pregnancy, labor, and delivery have resolved, and the body has reverted to the non-pregnant state [8,25,26]. In the fourth postpartum week, the body is still returning to the pre-pregnancy condition; therefore, we may have performed our evaluation too early to assess PPWR as a risk factor for GDM in the second pregnancy. Secondly, the previous study did not consider a family history of diabetes or previous GDM [8], although these are potential GDM risk factors [14–16]. Similar to previous investigations [14–16], our study revealed a significant association between a history of GDM and subsequent GDM. When interpreting the study results, the difference in explanatory variables between the previous study and ours needs to be considered.

Moreover, our multiple logistic regression analysis including PPWR assessed at the postpartum checkup as a continuous variable, showed no association between PPWR in the fourth week after delivery (adjusted OR, 1.18; 95% CI, 0.94–1.50) and GDM. As Liu et al. pointed out, there are only three times during a woman's life when she is almost certain to receive an examination from a trained health care provider: infancy, pregnancy, and the postpartum period, and if a chronic disease develops [8]. Given that most women will attend their postpartum appointment, we hoped to show an association between the PPWR at the time of this checkup and the risk of subsequent GDM to facilitate the identification of high-risk patients.

Returning to her pre-pregnancy weight is a challenge for many women, and previous studies demonstrated that 25% of women weigh $\geq$5 kg more than they did before pregnancy a year after delivery [27]. In this study, there was no significant difference in PPWR between obese and non-obese women (1.20±2.29 vs. 1.53±1.28 BMI unit, p = 0.51). Future research of the association between PPWR at the postpartum checkup and GDM in the next pregnancy must include a re-evaluation of the timing of data collection and the postpartum checkup, weight retention cut-off value. According to the current study results, attention should be paid to women with the traditional risk factors of obesity, the interpregnancy BMI change, prior history of GDM, and family history of GDM rather than PPWR–especially at four weeks after delivery.

Our study had limitations. Firstly, this was a single-center study, and it might be difficult to extrapolate our results to the general population. According to the sample size calculation method previously reported and widely used when performing logistic analysis [28–30], "Ten Events Per Variable" is a widely adopted minimal guideline criterion for performing logistic regression analysis. Based on this method, we needed at least 60 GDM women during the second pregnancy in this study. Since this study only included 42 GDM women during the second pregnancy, there is a possibility of the lack of or insufficient study power to detect a risk difference. Therefore, a large-scale, prospective, multicenter, cohort study is needed to confirm our results in the general population. Secondly, we did not evaluate some potential GDM risk factors, including a previous history of macrosomia [16] and polycystic ovary syndrome [31]. Thus, unmeasured confounders may be associated with GDM in the second pregnancy in this study. Thirdly, the generalizability of our findings may be limited by the homogeneity of our all-Japanese cohort.

Given the small sample size of our study and that our findings were inconsistent with those of Liu et al., we propose further investigation of the role of PPWR identified at the postpartum checkup in the pathophysiology of GDM in the second pregnancy in larger patient cohorts. If such a role exists, strategies should be developed to reduce the incidence of GDM and improve the perinatal prognosis in high-risk populations.

## Supporting information

**S1 Data.**
(XLSX)

## Acknowledgments

We thank the study subjects for the use of their personal data.

We would like to thank Editage (www.editage.com) for English language editing.

## Author Contributions

**Conceptualization:** Satoshi Shinohara, Motoi Takizawa.

**Data curation:** Satoshi Shinohara, Motoi Takizawa.

**Formal analysis:** Satoshi Shinohara.

**Investigation:** Satoshi Shinohara, Motoi Takizawa.

**Methodology:** Satoshi Shinohara.

**Project administration:** Satoshi Shinohara.

**Resources:** Satoshi Shinohara.

**Software:** Satoshi Shinohara.

**Validation:** Satoshi Shinohara.

**Visualization:** Satoshi Shinohara.

**Writing – original draft:** Satoshi Shinohara.

**Writing – review & editing:** Satoshi Shinohara, Atsuhito Amemiya, Motoi Takizawa.

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
