## [Decision Letter · Decision Letter 0]

2 Mar 2020

PONE-D-19-31440

Evaluation of weight retention four weeks after delivery as a risk factor for gestational diabetes mellitus in a subsequent pregnancy

PLOS ONE

Dear Dr Shinohara,

Thank you for submitting your manuscript to PLOS ONE. After careful consideration, we feel that it has merit but does not fully meet PLOS ONE’s publication criteria as it currently stands. Therefore, we invite you to submit a revised version of the manuscript that addresses the points raised during the review process.

We would appreciate receiving your revised manuscript by Apr 16 2020 11:59PM. To enhance the reproducibility of your results, we recommend that if applicable you deposit your laboratory protocols in protocols.io, where a protocol can be assigned its own identifier (DOI) such that it can be cited independently in the future. For instructions see: http://journals.plos.org/plosone/s/submission-guidelines#loc-laboratory-protocols

We look forward to receiving your revised manuscript.

Kind regards,

Zhong-Cheng Luo

Academic Editor

PLOS ONE

Additional Editor Comments:

1. Please add an ad hoc power calculaition for detecting a 2-times risk elevatation in GDM comparing postpartum BMI change >=1 (unstable) vs <1 (stable) in the Discussion. You observed almost 2 times risk elevation (adjusted OR=1.93), although it was not statistically significant. This may be due to the lack of or insufficient study power to detect a risk difference. If this is true, data interpretation should be adjusted for accordingly.

2. Please add a cloumn for the accurate P value to the precision of 3 decimal points for the adjusted OR in Table 2.

3. Line 184, why stated "(data not shown)" when you acturally presented the data?

'The funders had no role in study design, data collection and analysis, decision to publish, or preparation of the manuscript'

Please provide an amended Funding Statement that declares *all* the funding or sources of support received during this specific study (whether external or internal to your organization) as detailed online in our guide for authors at http://journals.plos.org/plosone/s/submit-now Please state what role the funders took in the study.  If any authors received a salary from any of your funders, please state which authors and which funder. If the funders had no role, please state: "The funders had no role in study design, data collection and analysis, decision to publish, or preparation of the manuscript."

Reviewers' comments:

Reviewer's Responses to Questions

**Comments to the Author**

1. Is the manuscript technically sound, and do the data support the conclusions?

Reviewer #1: Partly

Reviewer #2: Yes

2. Has the statistical analysis been performed appropriately and rigorously? 

Reviewer #1: I Don't Know

Reviewer #2: Yes

3. Have the authors made all data underlying the findings in their manuscript fully available?

Reviewer #1: Yes

Reviewer #2: Yes

4. Is the manuscript presented in an intelligible fashion and written in standard English?

Reviewer #1: Yes

Reviewer #2: Yes

5. Review Comments to the Author

Reviewer #1: The question about weight gain and retention and its impact on pregnancy outcomes is an important issue.

There a number of points that need clarification.

1. Was the pre-pregnancy weight for calculation of the BMI at the first visit self-reported?

2. How was the screening for GDM performed? Was it a 1-step or was there pre-screening with random BSL or GCT as both of these have been shown to miss up to 20% of cases of GDM? Understanding this is important for the non-Japanese reader.

You have made comment in your discussion that you were measuring weight retention at 4 weeks postpartum that may not accurately reflect the final post pregnancy return to the non-pregnant state.

It was not clear if you have made adjustment for the gestational length of the pregnancy as this may have a significant impact on the total weight gain and therefore the peak level from which the woman has to lose weight in returning to the non-pregnancy state.

Reviewer #2: In this paper, the authors analyzed 4 week postpartum weight retention (PPWR) using BMI in the 4th week after delivery in 566 nondiabetic, Japanese women with the risk for GDM in a subsequent pregnancy.

A Stable PPWR (4 wk BMI – 1st pregnancy prepregnancy BMI) < 1 BMI unit means that the individual lost almost all of their pregnancy weight gain by 4 weeks, correct? The definition of non-stable PPWR (gain of > 1 BMI unit) is only defined in the Abstract and should again be defined in the Methods and repeated in the results.

They found no difference in the risk for GDM according to their separation of groups according to BMI change at this 4 week follow-up visit, at which only about 1/3 of women had come back to < 1BMI difference from their prepregnancy BMI. They did find that obesity and interpregnancy weight gain, as expected, were risk factors for GDM. Could the authors analyze the PPWR among those who were obese compared to those not obese? Probably not, as the proportion who were obese was very low (71/566). The authors could mention in their Discussion, that given the current results, attention should be paid to women with the traditional risk factors of obesity, weight gain during pregnancy, prior history of GDM and family history of GDM rather than PPWR – especially at 4 weeks.

6. PLOS authors have the option to publish the peer review history of their article (what does this mean?). If published, this will include your full peer review and any attached files.

Reviewer #1: Yes: Jeremy J N Oats

Reviewer #2: No

---

## [Author Response · Author response to Decision Letter 0]

11 Mar 2020

March 5, 2020

Zhong-Cheng Luo

Academic Editor

PLOS ONE

Dear Editor:

I, along with my co-authors, would like to re-submit the attached manuscript, titled “Evaluation of weight retention four weeks after delivery as a risk factor for gestational diabetes mellitus in a subsequent pregnancy” as an original research article (manuscript ID: PONE-D-19-31440).

We appreciate your valuable comments, which have been very useful to us in improving the quality of our manuscript. All changes are shown in red in the revised manuscript. The entire manuscript has been also rechecked, and the necessary changes have been made in accordance with your suggestions. Our point-by-point responses to all of your comments have been prepared and provided below.

We believe that the changes made based on your comments have significantly improved our manuscript, and we hope that you will find it suitable for publication in PLOS ONE.

Sincerely,

Dr. Satoshi Shinohara

Department of Obstetrics and Gynecology

National Hospital Organization Kofu National Hospital

11-35 Tenjin, Kofu, Yamanashi 400-8533, Japan

Tel: +81-55-253-6131

Fax: +81-55-251-5597

Email: shinohara617@gmail.com

 

RESPONSES TO THE EDITOR’S COMMENTS

Comment 1

Please add an ad hoc power calculaition for detecting a 2-times risk elevatation in GDM comparing postpartum BMI change >=1 (unstable) vs <1 (stable) in the Discussion. You observed almost 2 times risk elevation (adjusted OR=1.93), although it was not statistically significant. This may be due to the lack of or insufficient study power to detect a risk difference. If this is true, data interpretation should be adjusted for accordingly.

Response 1

Thank you for your comment. We reviewed the sample size again according to your comments. When performing logistic analysis like this study, it seems that the sample size is often determined based on events per variable (EPV) rather than per ad hoc power calculation (A-C). Calculating the required sample size based on the previous report (10 EPV is a widely adopted minimal guideline criterion for performing logistic regression analysis), we needed at least 60 GDM women during the second pregnancy in this study. Therefore, since this study only included 42 GDM women during the second pregnancy, there is a possibility of the lack of or insufficient study power to detect a risk difference as pointed out. Therefore, we added in the limitation section; “According to the sample size calculation method widely used when performing logistic analysis, “Ten Events Per Variable” is a widely adopted minimal guideline criterion for performing logistic regression analysis. Based on this method, we needed at least 60 GDM women during the second pregnancy in this study. Since this study only included 42 GDM women during the second pregnancy, there is a possibility of the lack of or insufficient study power to detect a risk difference.”

A) Peduzzi P, Concato J, Kemper E, Holford TR, Feinstein AR. A simulation study of the number of events per variable in logistic regression analysis. J Clin Epidemiol. 1996;49: 1373-1379.

B) Moons KG, de Groot JA, Bouwmeester W, Vergouwe Y, Mallett S, Altman DG, et al. Critical appraisal and data extraction for systematic reviews of prediction modelling studies: the CHARMS checklist. PLoS Med. 2014;11: e1001744.

C) Moons KG, Altman DG, Reitsma JB, Ioannidis JP, Macaskill P, Steyerberg EW, et al. Transparent reporting of a multivariable prediction model for individual prognosis or diagnosis (TRIPOD): explanation and elaboration. Ann Intern Med. 2015;162: W1-W73.

Comment 2

Please add a cloumn for the accurate P value to the precision of 3 decimal points for

the adjusted OR in Table 2.

Response 2

Thank you for your comment. Following your comment, we revised Table 2.

Comment 3

Line 184, why stated "(data not shown)" when you acturally presented the data?

Response 3

Thank you for your comment. According to your comment, we added Table 3, and “Finally, a

multiple logistic regression model, into which we entered PPWR as a continuous variable,

was performed. After controlling for each variable, a history of previous GDM, and obesity

were significantly associated with GDM in the second pregnancy (history of previous GDM:

adjusted OR, 7.70; 95% CI, 3.08–19.2; obesity: adjusted OR, 4.13; 95% CI, 1.91–8.95) in

the multivariable logistic regression analysis. PPWR was not significantly associated with

GDM in the second pregnancy (adjusted OR, 1.19; 95% CI, 0.94–1.50) (Table 3).” in the

result. 

RESPONSES TO THE REVIEWER 1’S COMMENTS

Comment 1

Was the pre-pregnancy weight for calculation of the BMI at the first visit self-reported?

Response 1

Thank you for your comment. The pre-pregnancy weight for calculation of the BMI at the first 

visit was self-reported. We added, “The pre-pregnancy weight for calculation of the BMI at 

the first visit was self-reported.” in the data collection.

Comment 2

How was the screening for GDM performed? Was it a 1-step or was there pre-screening 

with random BSL or GCT as both of these have been shown to miss up to 20% of cases 

of GDM? Understanding this is important for the non-Japanese reader.

Response 2

Thank you for your comment. We added “In Japan, an initial screening test followed by the 

diagnostic oral glucose tolerance test (OGTT) in screen-positive women, known as the two-

step approach, is used for GDM diagnosis [16]. Moreover, the Japan Society of Obstetrics 

and Gynaecology has recommended the 50 g glucose challenge test (GCT) as a screening 

method for GDM [16]. Women with abnormal GCT results (serum glucose levels ≥140 mg/dL) 

subsequently undergo a 75-g OGTT for a definitive diagnosis.” in the data collection.

Comment 3

You have made comment in your discussion that you were measuring weight retention at 4 

weeks postpartum that may not accurately reflect the final post pregnancy return to the non-

pregnant state. It was not clear if you have made adjustment for the gestational length of the 

pregnancy as this may have a significant impact on the total weight gain and therefore the 

peak level from which the woman has to lose weight in returning to the non-pregnancy state.

Response 3

Thank you for your comment. Following your comment, we added an analysis of the gestational length of the first pregnancy in Table1. 

RESPONSES TO THE REVIEWER 2’S COMMENTS

Comment 1

A Stable PPWR (4 wk BMI – 1st pregnancy prepregnancy BMI) < 1 BMI unit means that the 

individual lost almost all of their pregnancy weight gain by 4 weeks, correct? The definition 

of non-stable PPWR (gain of > 1 BMI unit) is only defined in the Abstract and should again 

be defined in the Methods and repeated in the results.

Response 1

Thank you for your comment. According to your comment, we changed from “We classified 

women according to their postpartum check-up PPWR and defined a stable PPWR as ≤1 

BMI unit, per previous relevant publications” to “We classified women according to their 

postpartum check-up PPWR and defined a stable PPWR as ≤1 BMI unit and non-stable 

PPWR >1 BMI unit, per previous relevant publications” in the Methods. Moreover, we 

changed from “The characteristics of women with and without were similar except for greater 

first and second pregnancy, pre-pregnancy BMIs and lower first pregnancy, postpartum 

checkup BMIs in the stable PPWR group (Table 1).” to “The characteristics of women with a

stable (≤1 BMI unit) and a non-stable PPWR (>1 BMI unit) were similar except for greater 

first and second pregnancy, pre-pregnancy BMIs and lower first pregnancy, postpartum 

checkup BMIs in the stable PPWR group (Table 1).”

Comment 2

They found no difference in the risk for GDM according to their separation of groups 

according to BMI change at this 4 week follow-up visit, at which only about 1/3 of women had 

come back to < 1BMI difference from their prepregnancy BMI. They did find that obesity and 

interpregnancy weight gain, as expected, were risk factors for GDM. Could the authors 

analyze the PPWR among those who were obese compared to those not obese? Probably 

not, as the proportion who were obese was very low (71/566). The authors could mention in 

their Discussion, that given the current results, attention should be paid to women with the 

traditional risk factors of obesity, weight gain during pregnancy, prior history of GDM and 

family history of GDM rather than PPWR – especially at 4 weeks

Response 2

Thank you for your comment. According to your comment, we added “According to the 

current study results, attention should be paid to women with the traditional risk factors of 

obesity, the interpregnancy BMI change, prior history of GDM and family history of GDM 

rather than PPWR – especially at four weeks after delivery.” and “In this study, there was no 

significant difference in PPWR between obese and non-obese women (1.20±2.29 vs. 

1.53±1.28 BMI unit, p=0.51)” in the discussion.

---

## [Editor Report · Decision Letter 1]

16 Mar 2020

Evaluation of weight retention four weeks after delivery as a risk factor for gestational diabetes mellitus in a subsequent pregnancy

PONE-D-19-31440R1

Dear Dr. Shinohara,

We are pleased to inform you that your manuscript has been judged scientifically suitable for publication and will be formally accepted for publication once it complies with all outstanding technical requirements.

With kind regards,

Zhong-Cheng Luo

Academic Editor

PLOS ONE